# Favorable Effect of High-Density Lipoprotein Cholesterol on Gastric Cancer Mortality by Sex and Treatment Modality

**DOI:** 10.3390/cancers15092463

**Published:** 2023-04-25

**Authors:** Su Youn Nam, Seong Woo Jeon, Junwoo Jo, Oh Kyoung Kwon

**Affiliations:** 1Department of Internal Medicine, School of Medicine, Kyungpook National University, Daegu 41944, Republic of Korea; 2Department of Statistics, Kyungpook National University, Daegu 41944, Republic of Korea; 3Department of Surgery, Kyungpook National University Hospital, Daegu 41944, Republic of Korea

**Keywords:** gastric cancer, mortality, high-density lipoprotein cholesterol, sex, treatment modality

## Abstract

**Simple Summary:**

Studies on the effects of high-density lipoprotein cholesterol (HDL-C) on gastric cancer mortality are few, and the results are inconsistent. In this study, we investigated the effects of HDL-C on gastric cancer mortality and conducted sub-group analysis by sex and treatment modality. In this cohort study including patients with gastric cancer (*n* = 22,468), HDL-C was inversely related with mortality with a dose-dependent manner. In validation cohort (*n* = 3379), HDL-C was also inversely associated with mortality with a dose-dependent manner. The two cohorts demonstrated consistent favorable effects of higher HDL-C on mortality in both sexes. In validation cohort, the beneficial effect of HDL-C was observed in both gastrectomy and endoscopic resection (*p* for trend < 0.001) as more remarkable in endoscopic resection group. In this study, we explored that an increased HDL-C reduced mortality in both sexes and curative resection group.

**Abstract:**

Studies on the effects of high-density lipoprotein cholesterol (HDL-C) on gastric cancer mortality are few, and the results are inconsistent. In this study, we investigated the effects of HDL-C on gastric cancer mortality and conducted sub-group analysis by sex and treatment modality. Newly diagnosed patients with gastric cancer (*n* = 22,468) who underwent gastric cancer screening between January 2011 and December 2013 were included and followed up until 2018. A validation cohort (*n* = 3379) that had newly diagnosed gastric cancer from 2005 to 2013 at a university hospital, was followed up until 2017. HDL-C was inversely related with mortality; adjusted hazard ratio (aHR) 0.90 (95% confidence interval [CI], 0.83–0.98) for HDL-C of 40–49 mg/dL, 0.86 (0.79–0.93) for HDL-C of 50–59 mg/dL, 0.82 (0.74–0.90) for HDL-C of 60–69 mg/dL, and 0.78 (0.69–0.87) for HDL-C ≥ 70 mg/dL compared to HDL-C < 40 mg/dL. In the validation cohort, HDL-C was also inversely associated with mortality; aHR 0.81 (0.65–0.99) for HDL-C of 40–49 mg/dL, 0.64 (0.50–0.82) for HDL-C of 50–59 mg/dL, and 0.46 (0.34–0.62) for HDL-C ≥ 60 mg/dL compared to HDL-C < 40 mg/dL. The two cohorts demonstrated that higher HDL-C was associated with a low risk of mortality in both sexes. In validation cohort, this association was observed in both gastrectomy and endoscopic resection (*p* for trend < 0.001) as more remarkable in endoscopic resection group. In this study, we explored that an increased HDL-C reduced mortality in both sexes and curative resection group.

## 1. Introduction

Gastric cancer is the sixth common cancer and the second leading cause of cancer-related deaths worldwide [1]. High-density lipoprotein cholesterol (HDL-C) has been associated with a reduction in cardiovascular disease and related mortality [2,3]. This association may be related to the anti-atherosclerotic, anti-thrombotic, anti-inflammatory, anti-apoptotic, anti-oxidative, immune modulating, and endothelium protective effects of HDL-C [4,5]. Moreover, HDL-C has been reported to contribute to the development and prognosis of several cancers [6,7,8]. In European cohort study, high HDL-C levels were associated with a reduction in total cancer mortality [6]. HDL-C levels have shown an inverse association with breast [7] and gastric cancer [8]. In a recent cohort study, all-cause mortality and cancer mortality have an inverse association with HDL-C levels [9]. Low apolipoprotein A1 was associated with an increased risk of cancer mortality in patients who underwent percutaneous coronary intervention [10]. A recent meta-analysis also suggested that HDL-C is associated with mortality from all-cause, cardiovascular disease, and cancer [11]. 

The association between HDL-C level and prognosis in gastric cancer has been inconsistent in previous small studies. Normal preoperative HDL-C levels are reportedly associated with a better prognosis than low HDL-C levels in gastrectomy patients [12], whereas a low level of HDL-C in gastrectomy patients correlates with cancer progression but not survival in another study [13]. The association between serum HDL-C levels and overall gastric cancer mortality remains unknown. Furthermore, the effect of HDL-C level on mortality after endoscopic resection for early gastric cancers has not been reported. Previously, a sex-discrepant link between several cancer risks and body mass index (BMI) have been reported [14]. Recently, we showed a sex-discrepant effect of BMI on gastric cancer death in both hospital-based and nationwide studies [15,16]. However, sex-specific effect of HDL-C on gastric cancer death has not been reported.

In this cohort study, we used data from the National Health Insurance Service System (NHISS) to investigate the association of HDL-C levels with overall death risk in patients with gastric cancer. We validated our main results using data from a university hospital cohort. Furthermore, we analyzed the effects of HDL-C levels according to sex in both NHISS and validation cohorts. Finally, we analyzed the effects of HDL-C levels according to treatment modality (endoscopic resection and gastrectomy) in validation cohort.

## 2. Methods 

### 2.1. Data Extraction and Content

In this large population-based cohort study, we extracted data from the Korean NHISS (REQ0000045293), which included the National General Health Examination and National Cancer Screening Program (NCSP). Several studies on cancer using the data derived from NHISS have been published [17,18]. We extracted data on the patients’ sex, incomes, disease codes of the International Classification of Disease (10th revision) from T20 data (disease codes), treatment code for endoscopic and gastrectomy from T30 data (prescription and medical practice codes), presence of chronic disease, medication, smoking status (none, past, or current), alcohol consumption status, physical activity, family history, blood pressure, body mass index [BMI, weight/height^2^ (kg/m^2^)], lipid levels, and fasting glucose levels from participants’ questionnaires. After 12 h of fasting, blood glucose and lipid levels were measured. Moderate activity was defined as physical activity with light sweating for over 30 min per day. Alcohol drinking frequency was classified as none, 1/week, 2–3/week, 4–5/week, or ≥6/week. Moderate physical activity was categorized as none, 1–2 days/week, 3–5 days/week, or 6–7 days/week. Women factors included menopausal status (premenopausal, hysterectomy, and postmenopausal status), breastfeeding history (<6 months, ≥6 months and <1 year, ≥1 year, and never), delivery frequency (parity), use of oral contraceptives (never, <1 year, ≥1 year, and unknown), and estrogen replacement therapy (never, <2 years, ≥2 and <5 years, ≥5 years, and unknown). Serum HDL-C levels were measured in milligrams per deciliter (mg/dL). HDL-C level was classified as <40, 40–49, 50–59, 60–69, or ≥70 mg/dL. BMI was categorized as follows: Low (<18.5 kg/m^2^), normal (18.5–22.9 kg/m^2^), overweight (23–24.9 kg/m^2^), obesity I (25–29.9 kg/m^2^), or obesity II (≥30 kg/m^2^) [19]. The Korean NCSP provides endoscopy or gastrography for gastric cancer screening every other year for individuals over the age of 40 years [20].

### 2.2. Baseline Enrollment and Follow-Up in the NHISS Cohort

Patients with newly diagnosed gastric cancer who underwent NCSP between January 2011 and December 2013 were followed up until 2018. We excluded the patients who had been detected with gastric cancer before 2011 and those with pre-existing gastric cancer according to their responses to the NCSP questionnaires (Figure 1a). In this study, gastric cancer patients were defined as (1) patients who were diagnosed with gastric cancer by final histological diagnosis in NCSP or (2) patients who were suspected to have gastric cancer by final histological diagnosis in the NCSP, and then were confirmed as having gastric cancer (C16) by the T20 disease code within 6 months after NCSP. Relevant mortality data included death or survival status up to December 2018 and the month and year of death. The requirement for patient consent was waived since the NHISS provided raw data after the elimination of private information. This study was approved by an ethics committee called ‘the Institutional Review Board of Kyungpook National University Chilgok Hospital’ (KNUCH 2017-12-022).

### 2.3. Study Population and Follow-Up in the Validation Cohort

NHISS cohort includes many epidemiologic factors but does not include tumor characteristics and treatment. Therefore, we analyzed the hospital-based cohort to validate the association between HDL-C and the risk of death in gastric cancer patients considering tumor factors. Patients diagnosed with gastric cancer between 2005 and 2013 at Kyungpook National University Chilgok Hospital [KNUH] were included and followed up until 2017 [15]. Patients receiving treatment for other types of cancer at the time of diagnosis (*n* = 49), patients with remnant gastric cancer (*n* = 25), and those with other cancers appearing within 3 months (*n* = 13) were excluded (Figure 1b). Patients with an unknown prognosis (*n* = 3) and those without data on HDL-C levels at the time of diagnosis (*n* = 2628) were excluded. Tumor pathology was classified according to the World Health Organization (WHO) classifications. Tumor differentiation was classified as differentiated or undifferentiated. Gastric cancer sites were classified as distal third (antrum), middle third (lower body and mid body), or upper third (upper body, fundus, and cardia) of the stomach. Tumor stages were classified as I, II, III, and IV based on criteria from the American Joint Committee on Cancer’s staging system, seventh edition [21]. The final treatment method was classified as endoscopic submucosal dissection (ESD), curative gastrectomy, palliative chemotherapy, or palliative surgery. Patients were followed up until December 2017 and death information was acquired from the National Health Insurance Service records. Handling of missing data in KNUH cohort was provided in Appendix A. 

### 2.4. Statistical Analysis

Independent *t* tests or Pearson’s chi-squared were used to assess differences in demographic characteristics and HDL-C levels between the alive and dead groups. Survival of gastric cancer according to time (time to event) by HDL-C level was estimated using a Kaplan–Meier method. Proportional Hazards Assumption was observed (Appendix A). Mortality risk was measured with the hazard ratio (HR) and 95% confidence interval (CI) using the Cox proportional regression analysis. Significant variables in the unadjusted analyses were adjusted for in the multivariate analysis. A directed acyclic graph provides the assumed causal framework of covariate adjustments (Appendix A).

We also performed a sub-analysis according to sex and treatment methods, such as gastrectomy and endoscopic resection. We further adjusted for significant women factors in the women analysis. We further classified HDL-C levels to conduct a sensitivity analysis for the definition of exposure (Appendix A). HDL-C was classified by the National Cholesterol Education Program and Adult Treatment Panel (ATP III) [22]. HDL-C levels were further categorized into four groups. Statistical analyses were conducted using the SAS (version 9.4; SAS Institute, Cary, NC, USA). All statistical tests were two-sided, and statistical significance was set at *p* < 0.05.

## 3. Results

### 3.1. Baseline Characteristics of Patients

In the NHISS cohort, 22,468 patients diagnosed with gastric cancer were enrolled in the study, and 5086 deaths were detected until December 2018 (Figure 1a). Baseline demographic features (Table 1) and women factors (Appendix A) according to the HDL-C category were provided. Men comprised 65.9% of the patients, and the mean age was 64.6 years. The mean HDL-C level and BMI were 52.7 mg/dL and 23.8 kg/m^2^, respectively. The baseline demographic features and women factors by death and alive are also provided in Appendix A. Sex, baseline age, HDL-C, BMI, chronic diseases, such as hypertension, diabetes, heart disease, cerebrovascular disease, smoking status, drinking status, first-degree relative of gastric cancer, and physical activity were strongly associated with death. Women factors, including estrogen replacement therapy, breastfeeding history, and use of oral contraceptives, were also associated with death. 

Altogether, 3379 patients with gastric cancer in the KNUH cohort were eligible in the study (Figure 1b). We provided the baseline characteristics of included (presence of HDL-C levels) and excluded patients (absence of HDL-C levels) in Appendix A. The median follow-up period was 8 years. Men comprised 66.7% of the study population, and the mean age was 61.2 years. The mean HDL-C level and BMI were 48.4 mg/dL and 23.6 kg/m^2^, respectively. Gastric cancers were observed to occur in the lower (43.2%), middle (42.9%), and upper third (13.9%) of the stomach. Most gastric cancers were stage I (82.2%), whereas fewer cases were stage II (7.3%), III (5.9%), or IV (4.6%). Most patients underwent endoscopic submucosal dissection (27.3%) or gastrectomy (69.8%), whereas a few patients underwent palliative treatment (2.9%; Appendix A). 

### 3.2. Gastric Cancer Mortality According to HDL-C Levels in the NHISS Cohort

In the univariate analysis, many variables were linked with the death risk in gastric cancer patients (Appendix A). In the adjusted analysis, increased HDL-C levels reduced the risk of death in gastric cancer patients dose-dependently [adjusted HR (aHR) 0.90 for HDL-C levels of 40–49 mg/dL, 0.86 for HDL-C levels of 50–59 mg/dL, 0.82 for HDL-C levels of 60–69 mg/dL, and 0.78 for HDL-C levels ≥ 70 mg/dL compared to the group with the lowest levels (HDL-C < 40 mg/dL)]. Sub-analysis by sex also revealed that HDL-C consistently reduced the gastric cancer mortality dose-dependently in both sexes (Table 2).

Several other covariates were also associated with mortality in patients with gastric cancer (Table 2). Women had a lower risk of death than men. The direction of the effect of most covariates (age, house incomes, diabetes, physical activity, family history of gastric cancer, and use of lipid-lowering drug) on mortality was similar in both sexes. A lower BMI markedly increased gastric cancer mortality, and BMI values for the lowest risk of death were 25–29.9 kg/m^2^ in both sexes. Economic status was inversely correlated with mortality, and diabetes increased the risk of mortality in both sexes. Moderate physical activity, use of lipid-lowering drugs, and a first-degree relative of gastric cancer reduced the risk of mortality in both sexes. However, current smoking increased gastric cancer mortality in men, but not in women.

### 3.3. Gastric Cancer Mortality According to HDL-C Levels in the KNUH Cohort

In the adjusted analysis, an increase in HDL-C levels reduced the risk of gastric cancer mortality dose-dependently [aHR 0.86 for HDL-C levels of 40–49 mg/dL, 0.69 for HDL-C levels of 50–59 mg/dL, 0.54 for HDL-C levels of 60–69 mg/dL, and 0.42 for HDL-C levels ≥ 60 mg/dL compared to the group with the lowest level (HDL-C levels < 40 mg/dL)]. Sex-specific analysis also revealed that HDL-C consistently reduced the risk of gastric cancer mortality in both sexes (Table 3). Mortality was increased by 2.39-fold in patients with BMIs < 18.5 kg/m^2^ compared to those with BMIs of 25–29.9 kg/m^2^. Sub-analysis by sex revealed that the lowest BMI increased gastric cancer mortality in both sexes.

### 3.4. Sensitivity Analysis by Different HDL-C Levels

The risk of gastric cancer mortality according to different HDL-C definitions in the NHISS cohort is provided in Table 4. Normal HDL-C levels, according to the ATP III definition, were associated with a reduced risk of gastric cancer mortality compared to low HDL-C levels (aHR: 0.86 [total population], 0.88 [men], and 0.84 [women]). In the analysis by NECP classification, intermediate (aHR: 0.88 [total population], 0.90 [men], and 0.80 [women]) and high HDL-C levels (aHR: 0.82 [total population], 0.80 [men], and 0.87 [women]) were associated with a reduced risk of gastric cancer mortality compared to low HDL-C levels. Four categorical HDL-C levels demonstrated a dose-dependently reduced mortality risk according to an increase in HDL-C levels in the total population, men, and women.

The risk of gastric cancer mortality according to different HDL-C definitions in the KNUH cohort is presented in Table 4. Normal HDL-C levels, according to the ATP III definition, were associated with reduced gastric cancer mortality compared to low HDL-C levels (aHR: 0.72 [total population], 0.75 [men], and 0.56 [women]). In the analysis based on NECP classification, intermediate (aHR: 0.81 [total population], 0.82 [men], and 0.61 [women]) and high HDL-C (aHR: 0.52 [total population], 0.49 [men], and 0.52 [women]) levels were related with a reduced risk of gastric cancer mortality compared to low HDL-C levels. Four categorical HDL-C levels also demonstrated that high HDL-C levels reduced the mortality in the total population, men, and women.

### 3.5. Sub-Group Analysis by Treatment Method

The effect of HDL-C levels on gastric cancer mortality by treatment modality was investigated in KNUH cohort (Table 5). Normal HDL-C levels reduced the risk of mortality after gastrectomy (aHR, 0.84) and endoscopic resection (aHR, 0.63) compared to low HDL-C levels. In an analysis using four HDL-C classifications, an increase in HDL-C reduced mortality in gastric cancer patients with a dose-dependent manner in both gastrectomy and endoscopic resection (both *p* for trend < 0.001) and the beneficial effect of HDL-C was more remarkable in endoscopic resection group.

### 3.6. Sub-Group Analysis in Stage I

The effect of HDL-C levels on gastric cancer mortality in stage I was conducted since most of the cases were at stage I. Sensitivity analysis showed that the direction and size of the association between HDL-C and death risk in gastric cancer patients with stage I was similar with those in the overall stage in men, but the inverse association was stronger in women in the analysis of stage I cases than in the overall stage analysis. The effect direction of BMI on death risk (inverted J shape) was similar in both overall stage and stage I; however, the hazard ratio was larger in stage I analysis than in overall stage analysis (Appendix A). In sub-group analysis by treatment, the effect direction and size were similar in overall stage analysis and stage I analysis (Appendix A).

## 4. Discussion

In this large national cohort study, higher HDL-C levels reduced gastric cancer mortality dose-dependently. A favorable effect of HDL-C levels on gastric cancer mortality was consistently noted in both sexes. The validation cohort demonstrated similar results. The dose-dependent reduction in gastric cancer mortality by HDL-C levels was consistent after gastrectomy and endoscopic resection in validation cohort. The favorable effect of HDL-C levels was prominent in endoscopic resection group. In sensitivity analysis among stage I, the effect direction and size were similar with those in overall stage in men, but the effect size on women gastric cancer death is remarkable in stage I analysis than overall stage analysis.

In this study, higher HDL-C levels were associated with lower risk of mortality in a dose-dependent manner in the large NHISS (*n* = 22,518) and KNUH cohorts (*n* = 3379). Additionally, in the sensitivity analysis according to different definitions of HDL-C, both cohorts revealed that an increase in HDL-C levels was associated with lower risk of mortality in patients with gastric cancer. The association between HDL-C level and long-term mortality in patients with gastric cancer after gastrectomy was inconsistent in previous small studies. Furthermore, the effects of HDL-C level on mortality after endoscopic resection among patients with early gastric cancer have not been reported (Appendix A). A Japanese study (*n* = 184) suggested a better prognosis in normal preoperative HDL-C levels compared to low HDL-C levels [12]. A Chinese cohort study (*n* = 431) reported shorter overall survival (HR, 1.76) in patients with low postoperative HDL-C levels [23]. However, another Chinese cohort study (*n* = 258) revealed that a low HDL-C level after gastrectomy was associated with cancer progression but not patients’ survival [13]. This favorable effect of HDL-C was also reported in all-cause, cardiovascular, and overall cancer mortality [6,9,11]. The mechanism for the favorable outcome of high HDL-C level on death in gastric cancer patients is unclear. Anti-atherosclerotic, anti-thrombotic, anti-inflammatory, anti-apoptotic, anti-oxidative, immune modulating, and endothelium protective effects of HDL-C [4,5] may affect the recurrence or progression of cancer and the course of underlying diseases, such as cardiovascular and metabolic diseases.

To the best of our knowledge, this is the first study to analyze the association of HDL-C levels and gastric cancer mortality by sex and treatment modality. Sub-analysis by sex revealed a consistently favorable association between higher HDL-C and gastric cancer mortality in both sexes in the two cohorts. Sub-analysis by treatment revealed that the dose-dependent reduction in gastric cancer mortality by HDL-C levels was observed after gastrectomy and endoscopic resection in validation cohort. The favorable effect of HDL-C levels was prominent in endoscopic resection group. The KNUH cohort was adjusted for tumor factors, such as stage, location, and differentiation. The KNUH cohort, which estimated the risk of death from gastric cancer considering tumor factors, seems to be clinically relevant. Sensitivity analysis showed that the effect direction and size of HDL-C on death risk in gastric cancer patients with stage I were similar with those in overall stage in men, but the effect size on women gastric cancer death is remarkable in stage I analysis than overall stage analysis. In sub-group analysis by treatment, the effect direction and size were similar in overall stage analysis and stage I analysis.

In this study including two cohorts for gastric cancer, higher baseline HDL-C level was constantly associated with a reduced mortality regardless of sex or treatment modality. Therefore, the HDL-C level needs to be considered in the estimation of prognostic outcome in gastric cancer patients. In a model adjusted for tumor factors, such as stage, differentiation, and location, the HDL-C level has an inverse association with overall death. Therefore, the maintenance of normal or higher HDL-C needs to acquire a favorable outcome regardless of tumor factors or treatment methods in gastric cancer patients.

The two gastric cancer cohorts used in this study were similar in terms of sex distribution (male sex: 65.9 and 66.7% in the NHISS and KNUH cohorts, respectively) and BMI (23.8 kg/m^2^ and 23.6 kg/m^2^ in the NHISS and KNUH cohorts, respectively). The mean age was 64.6 and 61.2 years for the NHISS and KNUH cohorts, respectively. The HDL-C level was 52.7 and 48.4 mg/dL in the NHISS and KNUH cohorts, respectively. Gastric cancer derived from the NHISS cohort included gastric cancers identified through the NCSP in persons aged ≥40 years. Gastric cancers extracted from the KNUH cohort included gastric cancer through the NCSP, private cancer screening, or hospital examination due to warning signs covering all ages. 

In this study, mortality was the lowest among patients with BMIs of 25–29.9 kg/m^2^, and the highest mortality rate was observed in those with BMIs < 18.5 kg/m^2^ (aHR: 2.44 and 2.39 in the NHISS and KNUH cohorts, respectively). A lower BMI increased the gastric cancer mortality rate in both sexes in both cohorts. We previously revealed the hazardous effect of low BMI on gastric cancer mortality [16]. In a Korean Veterans Health Cohort study, gastric cancer mortality in men was the highest in underweight patients (HR 2.72 in BMI < 18.5 kg/m^2^) compared to those with BMIs of 25–27.4 kg/m^2^ [24]. In a women cohort study from the United Kingdom, low BMIs (<22.5 kg/m^2^) increased gastric cancer mortality by 1.47-fold [25]. 

Several covariates were associated with the mortality in patients with gastric cancer, and their associations were consistent with those reported in previous studies. The favorable outcome in women with gastric cancer (aHR, 0.73) is consistent with results from studies conducted in Korea and the United States [15,26]. Economic status being inversely correlated with mortality risk is consistent with results from previous studies [27]. The result that diabetes increased mortality risk (aHR, 1.21) is also compatible with results from a meta-analysis (summary risk ratio = 1.29) [28]. Current smoking increased the mortality risk in men with gastric cancer, but it had no effect on mortality in women. The finding that moderate physical activity reduced mortality dose-dependently accords with results from a meta-analysis [29]. A favorable outcome in patients with family history of gastric cancer (aHR, 0.82) in this study also accords with findings from a previous meta-analysis [30]. The favorable outcome of lipid-lowering drugs on cancer mortality (aHR, 0.79) was also consistent with previous studies [31,32,33,34,35].

This study has several strengths. First, this large sample size allowed us to evaluate the adjusted effects of HDL-C level on gastric cancer mortality. Next, we validated these associations in an independent cohort and obtained similar results. Third, this is the first investigation of the link between HDL-C level and gastric cancer mortality in an endoscopic resection group. Fourth, similar effects of covariates on mortality in patients with gastric cancer with the association demonstrated in previous studies support the reliability of this study. Finally, a sensitivity analysis conducted according to HDL-C classification in both the NHISS and KNUH cohorts confirmed the favorable role of higher HDL-C in gastric cancer death.

Nevertheless, this study has some limitations. First, the two cohorts were Korean. Therefore, future studies in other countries are necessary to confirm the relevance of our results. Second, detailed cancer information in the NHISS cohort is unknown despite it containing information on epidemiologic covariates. Therefore, we performed an adjusted analysis considering tumor factors using the KNUH cohort and produced similar results regarding the association with the NHISS cohort. Third, this study investigated the overall death risk not gastric cancer-specific death risk. 

## 5. Conclusions

High-serum HDL-C levels reduced gastric cancer mortality in the two independent cohorts. This favorable effect was also observed in an observed sub-analysis by sex. Moreover, the favorable effect of higher HDL-C levels was after gastrectomy and endoscopic resection. Therefore, this lipoprotein could be considered in the prognosis of gastric cancer. Further studies assessing the modification effect of HDL-C on gastric cancer mortality are warranted to elucidate the factors that could reduce gastric cancer mortality.

## Figures and Tables

**Figure 1 cancers-15-02463-f001:**
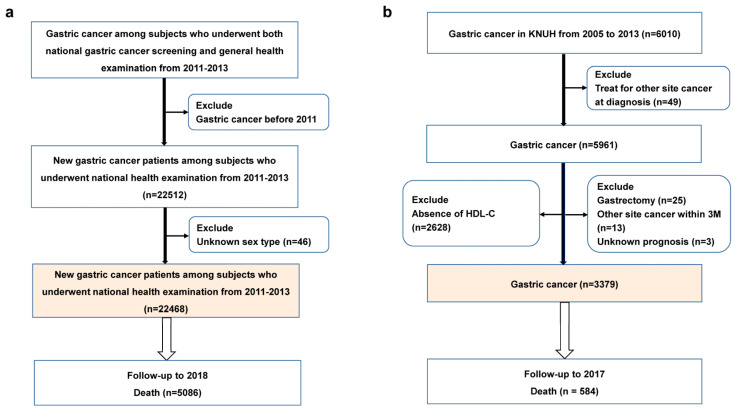
Study flowchart (**a**) The National Health Insurance Service System (NHISS) cohort. Participants who underwent national gastric cancer screening and the National General Health Examination protocol were surveyed to identify newly detected patients with gastric cancer. In total, 22,468 patients with gastric cancer were deemed eligible. (**b**) The Kyungpook National University Hospital (KNUH) cohort. Patients with gastric cancer who were newly diagnosed at the KNUH were enrolled. In total, 3379 patients with primary gastric cancer were deemed eligible.

**Table 1 cancers-15-02463-t001:** Baseline characteristics of gastric cancer patients by baseline HDL-C (NHISS cohort).

	HDL-C, mg/dL				
	<40 (*n* = 3562)	40–49 (*n* = 6970)	50–59 (*n* = 6133)	60–69 (*n* = 3552)	≥70 (*n* = 2251)
Person years	22,382.0	45,223.1	40,267.9	23,513.6	14,958.1
Men, no (%)	2840 (80.0)	5050 (72.7)	3920 (64.2)	2130 (60.2)	1280 (57.2)
Age, mean (SD)	65.9 (9.8)	64.8 (9.7)	64.5 (9.8)	64.1 (10.0)	63.5 (10.1)
Economic status, mean (SD) ^†^	12.3 (5.8)	12.4 (5.8)	12.2 (5.9)	12.1 (5.8)	12.0 (5.8)
Incomes, median (IQR) ^†^	14 (8, 17)	14 (8, 17)	13 (7, 17)	13 (7, 17)	13 (7, 17)
HDL-C, mg/dL, median (IQR)	36 (32,38)	45 (42, 47)	54 (52, 56)	64 (61, 66)	76 (73, 83)
BMI, kg/m^2^, median (IQR)	25 (23, 26)	24 (22, 26)	24 (22, 26)	23 (21, 25)	23 (21, 25)
BMI, kg/m^2^, no (%)					
<18.5	50 (1.6)	160 (2.3)	200 (3.4)	140 (4.0)	130 (6.1)
18.5–22.4	970 (27.4)	2190 (31.5)	2290 (37.5)	1500 (42.5)	1120 (50.0)
22.5–24.9	990 (27.9)	1910 (27.5)	1620 (26.5)	920 (26.1)	490 (22.0)
25–29.9	1400 (39.5)	2460 (35.4)	1830 (30.0)	890 (25.3)	450 (20.2)
≥30	130 (3.7)	220 (3.2)	160 (2.6)	70 (2.1)	30 (1.6)
Hypertension, no (%)	1600 (45.1)	2950 (42.4)	2430 (39.8)	1320 (37.3)	790 (35.4)
Heart disease, no (%)	210 (6.0)	360 (5.2)	270 (4.4)	140 (4.0)	70 (3.3)
Diabetes mellitus, no (%)	880 (24.8)	1290 (18.6)	850 (13.9)	420 (11.9)	220 (10.2)
Cerebrovascular disease, no (%)	100 (2.9)	160 (2.3)	120 (2.0)	50 (1.7)	20 (1.2)
Lipid-lowering drug, no (%)	110 (3.2)	250 (3.6)	260 (4.4)	150 (4.3)	90 (4.3)
Smoking status, no (%)					
Never	1590 (44.8)	3400 (48.9)	3340 (54.5)	2070 (58.4)	1360 (60.4)
Past	990 (28.0)	1920 (27.7)	1530 (25.0)	840 (23.7)	430 (19.5)
Current	970 (27.3)	1620 (23.4)	1250 (20.4)	630 (17.9)	450 (20.1)
Drinking frequency, no (%)					
None	2280 (64.3)	4120 (29.3)	3550 (57.9)	1980 (55.8)	1110 (49.7)
1/week	750 (21.3)	1610 (23.2)	1330 (21.9)	750 (21.2)	460 (20.4)
2–3/week	290 (8.4)	690 (10.0)	690 (11.4)	440 (12.4)	300 (13.7)
4–5/week	100 (3.0)	270 (4.0)	260 (4.3)	200 (5.7)	170 (7.8)
≥6/week	110 (3.1)	240 (3.5)	280 (4.6)	170 (4.9)	180 (8.4)
Family History of GC, no (%)	400 (11.5)	870 (12.6)	740 (12.3)	490 (13.9)	310 (14.0)
Moderate activity, no (%) ^‡^					
None	2240 (63.2)	4310 (61.9)	3730 (60.9)	2170 (61.3)	1340 (59.6)
1–2 days/week	560 (15.8)	1170 (16.8)	1010 (16.5)	580 (16.4)	380 (17.2)
3–5 days/week	510 (14.4)	990 (14.3)	970 (15.9)	550 (15.5)	350 (15.6)
6–7 days/week	230 (6.6)	480 (6.9)	400 (6.7)	240 (6.9)	170 (7.6)

^†^ Economic status refers to 20th percentile of income (score 1–20). 1 means the lowest 5% and 20 is the highest 5%. ^‡^ Moderate activity means physical activity with light sweating over 30 min in a day. For example, brisk walking, jogging, swimming leisurely, bicycling—light effort, gardening, and some housework, such as vacuuming. BMI, body mass index; DM, diabetes mellitus; GC, gastric cancer; HDL-C, high-density lipoprotein cholesterol; SD, standard deviation.

**Table 2 cancers-15-02463-t002:** Adjusted hazard ratio for gastric cancer death in NHISS cohort.

	Total *		Men **		Women ^†^	
	aHR (95% CI)	*p*-Value	aHR (95% CI)	*p*-Value	aHR (95% CI)	*p*-Value
HDL-C, mg/dL						
<40	1		1		1	
40–49	0.90 (0.83–0.98)	0.011	0.91 (0.83–0.99)	0.035	0.86 (0.72–1.03)	0.109
50–59	0.86 (0.79–0.93)	0.001	0.90 (0.81–0.99)	0.030	0.74 (0.62–0.89)	0.002
60–69	0.82 (0.74–0.90)	<0.001	0.80 (0.71–0.90)	0.001	0.81 (0.66–0.99)	0.042
≥70	0.78 (0.69–0.87)	<0.001	0.80 (0.70–0.92)	0.001	0.69 (0.55–0.87)	0.002
Women	0.73 (0.68–0.79)	<0.001				
Age, yr	1.07 (1.07–1.07)	<0.001	1.08 (1.07–1.08)	<0.001	1.07 (1.06–1.08)	<0.001
Incomes	0.99 (0.98–0.99)	<0.001	0.98 (0.98–0.99)	<0.001	0.99 (0.98–1.00)	0.023
BMI, kg/m^2^						
<18.5	2.44 (2.14–2.78)	<0.001	2.55 (2.20–2.96)	<0.001	2.12 (1.61–2.78)	<0.001
18.5–22.4	1.48 (1.37–1.59)	<0.001	1.47 (1.34–1.60)	<0.001	1.45 (1.26–1.67)	<0.001
22.5–24.9	1.17 (1.08–1.26)	0.001	1.18 (1.07–1.30)	0.001	1.12 (0.95–1.31)	0.177
25–29.9	1		1		1	
≥30	1.07 (0.87–1.30)	0.540	1.10 (0.84–1.43)	0.499	1.04 (0.76–1.42)	0.808
Hypertension	0.92 (0.87–0.98)	0.009	0.91 (0.85–0.98)	0.008	0.98 (0.86–1.10)	0.685
Heart disease	1.08 (0.96–1.21)	0.197	1.01 (0.88–1.16)	0.844	1.24 (1.00–1.54)	0.048
Diabetes	1.21 (1.13–1.30)	<0.001	1.22 (1.13–1.33)	<0.001	1.20 (1.04–1.39)	0.014
Stroke	1.15 (0.98–1.34)	0.091	1.21 (1.01–1.43)	0.037	0.94 (0.65–1.35)	0.724
Moderate activity						
None	1		1		1	
1–2 days/week	0.88 (0.81–0.96)	0.004	0.91 (0.83–1.00)	0.053	0.81 (0.68–0.97)	0.019
3–5 days/week	0.75 (0.69–0.83)	<0.001	0.76 (0.68–0.84)	<0.001	0.76 (0.62–0.93)	0.007
6–7 days/week	0.80 (0.71–0.89)	0.001	0.83 (0.73–0.94)	0.003	0.66 (0.49–0.90)	0.009
Smoking status						
Never	1	1	1		1	
Past	0.97 (0.90–1.05)	0.451	0.98 (0.91–1.07)	0.691	0.82 (0.49–1.38)	0.459
Current	1.12 (1.03–1.21)	0.008	1.14 (1.05–1.25)	0.002	1.06 (0.73–1.54)	0.752
Drinking frequency						
None	1		1		1	
1/week	0.87 (0.80–0.94)	0.001	0.86 (0.79–0.93)	0.001	1.00 (0.80–1.25)	0.994
2–3/week	0.80 (0.72–0.89)	<0.001	0.82 (0.73–0.91)	0.001	0.55 (0.29–1.04)	0.065
4–5/week	1.02 (0.89–1.16)	0.798	1.02 (0.89–1.17)	0.827	1.26 (0.52–3.04)	0.615
≥6/week	1.02 (0.90–1.15)	0.782	1.03 (0.90–1.17)	0.704	0.18 (0.03–1.24)	0.082
Family History of GC	0.82 (0.74–0.90)	<0.001	0.83 (0.74–0.92)	0.001	0.81 (0.67–0.97)	0.021
Lipid-lowering drug	0.79 (0.67–0.93)	0.004	0.88 (0.72–1.07)	0.196	0.68 (0.51–0.92)	0.013

* Adjusted for age, sex, incomes, hypertension, diabetes, cerebrovascular disease, heart disease, smoking status, drinking frequency, body mass index, high-density lipoprotein, lipid-lowering drug, physical activity, and family history of gastric cancer. ** Adjusted for age, incomes, hypertension, diabetes, cerebrovascular disease, heart disease, smoking status, drinking frequency, body mass index, high-density lipoprotein, lipid-lowering drug, physical activity, and family history of gastric cancer. ^†^ Adjusted for age, incomes, hypertension, diabetes, cerebrovascular disease, heart disease, smoking status, drinking frequency, body mass index, high-density lipoprotein, lipid-lowering drug, physical activity, family history of gastric cancer, menopausal status, estrogen replacement therapy, breastfeeding duration, and oral pill. aHR, adjusted hazard ratio; BMI, body mass index; CI, confidence interval; HDL-C, high-density lipoprotein cholesterol.

**Table 3 cancers-15-02463-t003:** Hazard ratio for gastric cancer death in KNUH cohort (validation cohort).

		Unadjusted Analysis		Adjusted Analysis					
		Total		Total (*n* = 3379)		Men (*n* = 2252)		Women (*n* = 1126)	
	Event/Total No	HR (95% CI)	*p*	aHR (95% CI)	*p* *	aHR (95% CI)	*p* ^†^	aHR (95% CI)	*p* ^†^
HDL-C, mg/dL									
<40	235/1006	1		1		1		1	
40–49	170/1001	0.67 (0.55–0.81)	<0.001	0.86 (0.69–1.07)	0.185	0.91 (0.71–1.17)	0.461	0.84 (0.52–1.36)	0.477
50–59	109/719	0.58 (0.46–0.72)	<0.001	0.69 (0.54–0.89)	0.004	0.72 (0.54–0.96)	0.024	0.55 (0.33–0.94)	0.027
60–69	44/359	0.44 (0.32–0.61)	<0.001	0.54 (0.38–0.77)	0.0006	0.54 (0.35–0.83)	0.005	0.52 (0.28–0.95)	0.035
≥70	26/294	0.30 (0.20–0.45)	<0.001	0.42 (0.27–0.66)	0.0002	0.42 (0.24–0.75)	0.003	0.40 (0.18–0.87)	0.021
BMI, kg/m^2^									
<18.5	59/138	4.11 (3.00–5.58)	<0.001	2.39 (1.73–3.29)	<0.001	2.45 (1.67–3.56)	<0.001	1.95 (1.02–3.78)	0.046
18.5–22.9	233/1276	1.42 (1.15–1.76)	0.001	1.35 (1.09–1.69)	0.006	1.36 (1.06–1.77)	0.016	1.22 (0.77–1.95)	0.397
23–24.9	140/898	1.16 (0.91–1.47)	0.224	1.16 (0.91–1.47)	0.233	1.24 (0.94–1.63)	0.133	1	
25–29.9	132/968	1		1		1		1.05 (0.63–1.74)	0.864
≥30	13/69	1.48 (0.84–2.61)	0.179	1.74 (0.98–3.08)	0.059	2.53 (1.31–4.88)	0.006	0.96 (0.29–3.21)	0.944

* Adjusted for age, sex, body mass index, HDL, diabetes, tumor location, stage, differentiation, and treatment method. ^†^ Adjusted for age, body mass index, HDL, diabetes, tumor location, stage, differentiation, and treatment method. aHR, adjusted hazard ratio; BMI, body mass index; CI, confidence interval; HDL-C, high-density lipoprotein cholesterol; HR, hazard ratio.

**Table 4 cancers-15-02463-t004:** Adjusted hazard ratio for gastric cancer death by different HDL-C definitions in the NHISS and KNUH cohorts.

		Unadjusted		Adjusted					
	Event/Total			Total		Men		Women	
NHISS Cohort		aHR (95% CI)	*p*-Value	aHR (95% CI)	*p*-Value *	aHR (95% CI)	*p*-Value **	aHR (95% CI)	*p*-Value ^†^
HDL-C by ATP III									
Low	1337/5457	1		1		1		1	
Normal	3745/16965	0.88 (0.83–0.94)	<0.001	0.86 (0.81–0.92)	<0.001	0.88 (0.81–0.95)	0.002	0.84 (0.74–0.95)	0.004
HDL-C by NECP									
Low	1338/5462	1		1		1		1	
Intermediate	2563/11174	0.92 (0.86–0.99)	0.018	0.88 (0.83–0.95)	0.001	0.90 (0.83–0.98)	0.017	0.80 (0.69–0.93)	0.004
High (≥60)	1184/5803	0.81 (0.75–0.88)	<0.001	0.82 (0.75–0.89)	<0.001	0.80 (0.72–0.89)	<0.001	0.87 (0.75–1.01)	0.068
HDL-C by 4 groups									
Very low (<30)	132/377	1		1		1		1	
Low	1206/5082	0.62 (0.51–0.74)	<0.001	0.68 (0.57–0.82)	<0.001	0.68 (0.55–0.84)	0.001	0.65 (0.42–1.00)	0.051
Intermediate	2563/11174	0.59 (0.49–0.71)	<0.001	0.62 (0.52–0.75)	<0.001	0.65 (0.53–0.79)	<0.001	0.53 (0.34–0.82)	0.005
High (≥60)	1184/5803	0.52 (0.43–0.62)	<0.001	0.58 (0.48–0.69)	<0.001	0.57 (0.46–0.70)	<0.001	0.58 (0.37–0.89)	0.014
KNUH cohort	Event/total	aHR (95% CI)	*p*-value	aHR (95% CI)	*p*-value ^§^	aHR (95% CI)	*p*-value ^€^	aHR (95% CI)	*p*-value ^€^
HDL-C by ATP III									
Low	282/1326	1		1		1		1	
Normal	302/2053	0.62 (0.53–0.73)	<0.001	0.72 (0.6–0.86)	<0.001	0.75 (0.6–0.92)	0.007	0.56 (0.39–0.83)	0.003
HDL-C by NECP									
Low	282/1326	1		1		1		1	
Intermediate	232/1400	0.72 (0.61–0.86)	<0.001	0.81 (0.67–0.99)	0.035	0.82 (0.66–1.03)	0.088	0.61 (0.38–0.97)	0.035
High (≥60)	70/653	0.42 (0.33–0.55)	<0.001	0.52 (0.39–0.69)	<0.001	0.49 (0.34–0.71)	<0.001	0.52 (0.33–0.84)	0.007
HDL-C by 4 groups									
Very low (<30)	82/244	1		1		1		1	
Low	200/1082	0.52 (0.4–0.67)	<0.001	1.04 (0.78–1.4)	0.784	1.38 (0.97–1.96)	0.072	0.66 (0.37–1.18)	0.163
Intermediate	232/1400	0.44 (0.34–0.56)	<0.001	0.83 (0.63–1.11)	0.208	1.02 (0.74–1.42)	0.901	0.44 (0.24–0.83)	0.011
High (≥60)	70/653	0.26 (0.19–0.35)	<0.001	0.53 (0.37–0.76)	<0.001	0.61 (0.39–0.94)	0.025	0.37 (0.19–0.72)	0.003

* Adjusted for age, sex, income, hypertension, diabetes, cerebrovascular disease, heart disease, body mass index, physical activity, smoking, drinking, lipid-lowering drug, and family history. ** Adjusted for age, hypertension, diabetes, cerebrovascular disease, heart disease, body mass index, physical activity, smoking, drinking, lipid-lowering drug, and family history. ^†^ Adjusted for age, incomes, hypertension, diabetes, cerebrovascular disease, heart disease, smoking status, drinking frequency, body mass index, high-density lipoprotein, physical activity, family history of gastric cancer, menopausal status, estrogen replacement therapy, breastfeeding duration, and oral pill. ^§^ Adjusted for age, sex, fasting glucose, stage, location, differentiation, and treatment method. ^€^ Adjusted for age, fasting glucose, stage, location, differentiation, and treatment method. aHR, adjusted hazard ratio; BMI, body mass index; CI, confidence interval; HDL-C, high-density lipoprotein cholesterol. Definition of HDL-C was provided in Appendix A.

**Table 5 cancers-15-02463-t005:** Hazard ratio for gastric cancer death by treatment in KNUH cohort.

	KNUH					
	Gastrectomy (*n* = 2358)			ESD (*n* = 922)		
	HR (95% CI) *	*p*-Value	*p* for Trend	HR (95% CI) *	*p*-Value	*p* for Trend
HDL-C by ATP III						
Low	1			1		
Normal	0.63 (0.51–0.78)	<0.001		0.59 (0.39–0.88)	0.011	
HDL-C by 4 groups						
Very low (<30)	1			1		
Low	0.90 (0.64–1.27)	0.547		0.68 (0.36–1.30)	0.241	
Intermediate	0.63 (0.46–0.87)	0.004		0.48 (0.24–0.94)	0.031	
High (≥60)	0.44 (0.30–0.64)	<0.001	<0.001	0.06 (0.01–0.50)	0.009	<0.001

* Adjusted for age, sex, BMI, fasting glucose, stage, location, and differentiation. aHR, adjusted hazard ratio; BMI, body mass index; CI, confidence interval; ESD, endoscopic submucosal dissection; HDL-C, high-density lipoprotein cholesterol. Definition of HDL-C was provided in Appendix A.

## Data Availability

NHISS data are not available and KNUH data are available from the corresponding author on reasonable request.

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
