# Peer review of "Favorable Effect of High-Density Lipoprotein Cholesterol on Gastric Cancer Mortality by Sex and Treatment Modality"

_cancers, 2023, doi:10.3390/cancers15092463_

Round 1
Reviewer 1 Report
Thank you for giving me the opportunity to review this manuscript. Overall, the manuscript is well written, but I suggest the following comments for the authors to consider prior to publication:
1. The introduction is rather short and does not entirely captures the burden of the problem and the critic of the literature, coherently it doesn't highlight the needs gaps of the current study.
2. Under methodology part, can the authors provide a causal diagram (e.g., directed acyclic graphs) on the factors related to the outcome variable? It will be more meaningful to understand the relationships between variables before the results part is interpreted.
3. It is not necessary to report p-value in such circumstances. In such studies that determines the hard endpoints, it will be important to interpret the effect size and effect estimates (95% CI) to be sufficient.
4. Under discussion, it would be good to discuss (with a sub-section) the implications of the study to clinical practice.
Author Response
We attached the response file.

Reviewer 2 Report
This manuscript presents the results of two studies showing that higher HDLc levels are associated with lower mortality among people diagnosed with gastric cancer in two studies based on data from national health examination and screening programs for gastric cancer. This is a novelty result of scientific interest that deserves to be further investigated and reproduced in other populations. However, the manuscript has some formal flaws and several methodological concerns that need to be clarified.
One main concern is related to when the HDLc and other variables were measured. Were they collected at baseline? Would it be possible they were altered by the cancer process and thus, a causing reverse causation.
There are other concerns in relation to study design, analysis and the written manuscript that need to be reviewed in depth (see specific comments to authors). In this survival study, one would expect some figures as Kaplan-Meier survival curves by categories of HDLc levels.
Specific comments to authors
Introduction
1. The introduction is too brief with a poor description of the literature findings in relation to HDLs mechanisms that could be relevant for cancer risk, associations with other cancers, etc.
2. Line 30. The use of the term “reduces” is inappropriate. I suggest “…has been associated with…“, instead.
3. The study aim should be more precisely defined, providing some detail of the setting, study design, follow-up, meaning of terms (eg. mortality), etc. It seems that there are three study aims. The meaning of second aim named validation of the results is unclear, and the third is in fact, a stratified analysis by two variables (sex and treatment type).
4. The use of the term “effect” is inappropriate. Association should be used instead.
Methods
5. line 50. “Many studies on cancer using the data derived from NHISS have been published.11,12” Although, only two references were included, one on thyroid cancer and other for pancreas cancer. This should be better documented.
5. Lines 52-53. Please, clarify the meaning of T20 and T30 data.
6. lines 64-69. The selection of cases is unclear. Apparently, only those newly diagnosed between 2011-13 in the national screening program. However, the sentence “The patients who had been detected with gastric cancer before 2011 and those with pre-existing gastric cancer according to their responses to the NCSP questionnaires (Figure 1A).” is confusing. Were these cases also included?
More detailed information should be provided for the National Health Insurance system (NHIS) cohort study, for screening and health examination survery: when the program was started? the total number of screened people, how a new diagnosis of gastric cancer was confirmed, etc. Some additional data on the health problem that the gastric cancer causes in Korea would be also of interest.
7. The aim of the validation study is unclear. Validation of what? Patients from this study were diagnosed in a different period. Detailed information on this study should be provided.
8. The number of patients excluded due to the lack of data on HDL-C was too high. Were these patients different from those with n HDL-c data?
9. Is the classification of tumor differentiation based in any international system?
10. Tumor stages were classified as I (82%), II, III, and IV based on criteria from the American Joint Committee on Cancer’s staging system, seventh edition. Is there any reference for this system? It is to note that most of cases were at stage I. This would require a sensitive analysis including only these cases.
11. line 92. Please provide a better explanation for the treatment option: “or palliative OP/open and closed”.
12. The definition of the study variables should be placed in a specific section (eg. covariables), better than in statistical analysis.
13. Line 101. Please clarify the definition and criteria for categories of alcohol consumption and physical activity, and other variables. Details of the categories should be also presented in footnotes of tables 3-5.
14. The statistical analysis should be better justified. The definition of the time variable and the assumptions of the Cox models should be considered. Survival curves should be presented.
15. Lines 110-111. The sentence is for HDL-C is confusing. Five categories were defined in the previous paragraph.
16. Is there any justification for using two different software for survival analysis?
Results
17. Since HDL-c values are higher in women than in men, separate analysis is well justified.
18. Line 125. When described in the text, the associations are not well defined. Were they from univariate analyses?
19. line 134. Is there information on the HDLc levels by tumor stage?. Since most of gastric cancer were diagnosed at stage I (82%), it would be of interest to assess survival using these patients only, just in case that more advances cancer stages cause lower levels of HDLs (reverse causation). A sensitivity analysis using only information for Stage I patients would be of interest.
20. line 167. “A lower BMI markedly increased gastric cancer mortality” Could this association be due to the cancer process?
21. Table 5. Please, use the complete name in footnote of the table 5 for the (ESD), endoscopic submucosal dissection.
22. What are the main causes of death for gastric cancer patients in the study?.
Discussion.
The discussion is poorly developed. No comments on the potential mechanisms for this novelty association. Is there any literature for other cancers?
23. Line 234. Please, use was associated with lower risk of mortality, instead of reduces.
25. Line 253. The link. Use the association instead.
24. Line 259. “The KNUH cohort was adjusted for tumor factors such as stage, location, and differentiation. The KNUH cohort, which estimated the risk of death from gastric cancer considering tumor factors, seems to be clinically relevant.” It would be of interest a sensitivity analysis only with stage I patients.
26. Helicobacter pilori is a relevant causal factor for gastric cancer. Is there any information ion this important factor?
Conclusion
25. The conclusion should be placed in the corresponding section according to the editorial recommendation.
Author Response
We attached the response file.

Reviewer 3 Report
The manuscript Favorable effect of high-density lipoprotein cholesterol on gas tric cancer mortality by sex and treatment modality was presented for the peer review. HDL-C has been reported to contribute to the development and prognosis of several cancers. Authors investigated the effects of HDL- C on gastric cancer mortality and conducted sub-group analysis by sex and treatment modality. Methods are sound and relevant.
I have several suggestions.
Line 165 p5 The phrase The direction of the effect of most covariates on mortality was similar in both sexes needs detailed information. Please point out these covariates in the text.
Please clear your sentence Normal HDL-C levels reduced the risk of mortality after gastrectomy (aHR, 0.84) and endoscopic resection (aHR, 0.63) compared to low HDL-C levels. I mean possible mechanistic link between HDL-C level and risk of mortality.
Author Response
We attached the response file.

Reviewer 4 Report
Review for the manuscript Favorable effect of high-density lipoprotein cholesterol on gastric cancer mortality by sex and treatment modality submitted to CANCER.
Dear Editor and authors, thank you for the opportunity to review this very interesting manuscript. After careful evaluation, I suggest some modifications before it can be published.
Overall comments:
This is a very interesting manuscript.
Please review the text. There are some punctuation errors in the text. For example, some commas are missing.
ABSTRACT
In the first sentence of this section we find "Studies on the effects of high-density lipoprotein cholesterol (HDL-C) on gastric cancer 9 mortality are few, and the results are inconsistent." I suggest modifying for "Studies on the effects of high-density lipoprotein cholesterol (HDL-c) on gastric cancer mortality are few and the results are inconsistent."
Please, change "HDL-C" for "HDL-c" along with the text.
Although HDL is a well-known acronym, I suggest defining it the first time it appears in the Abstract and in the text.
In lines 20-22 we can read "The two cohorts demonstrated consistent favorable effects of higher HDL-C on mortality in both sexes. In validation cohort, the beneficial effect of HDL-C was observed in both gastrectomy and endoscopic resection …). What are the favorable effects of HDL-c? I think the reader would like to see this in the Abstract. Moreover, I suggest change for "in the validation cohort, the beneficial..."
KEYWORDS
The keywords are adequate.
INTRODUCTION
The Introduction is very concise. I think that in this section, the authors could explore a little more about gastric cancer. It would be interesting to see numbers in epidemiology and not just know that it is an important cause of death. What are the main causes of this type of cancer?
There are a plethora of new articles on this type of cancer (epidemiology and general aspects). Please, include references published in 2022 and 2023 in this section. Please consult PUBMED.com
METHODS
This section was adequately described. However, we can find
RESULTS
Please, correct “HDL-C, high density lipoprotein” for “HDL-c, high-density lipoprotein” in all the tables. Please, also check these tables for other similar mistakes.
Tables:
Supplementary Table 1.
Supplementary Table 1. Five different definition of HDL-C categories. Please, change for “Supplementary Table 1. Five different definitions of HDL-C categories”
In the legends, change “HDL-C, high density lipoprotein cholesterol” For “HDL-c, high-density lipoprotein cholesterol.”
When we see “NECP”, would it be “NCEP”?
If you mean “The Third Report of The National Cholesterol Education Program”, please, include in the legend of the table.
Supplementary Table 2.
Sometimes the authors use “post-menopause” and sometimes“postmenopause!
Please, change “Breast feeding duration” for “Breastfeeding duration”
Supplementary Table 3.
Please, change “Breast feeding duration” for “Breastfeeding duration”
Please, change “
|
1-2 day/week |
|
3-5 day/week |
For:
|
1-2 days/week |
|
3-5 days/week |
Please change “*p-values are derived from t-test or chi-square test. †Economic status refers 20 tile of income (score 1-20). 1 means lowest 5% and 20 is highest 5%. ‡Moderate activity means physical activity with light sweating over 30 minutes in a day. For examples, brisk walking, jogging, swimming leisurely, bicycling light effort, gardening, and some housework such as vacuuming.
BMI, body mass index; DM, diabetes mellitus; GC; gastric cancer; HDL-C, high density lipoprotein cholesterol; SD, standard deviation.” For:
“*p-values are derived from t-test or chi-square test. †Economic status refers 20 tile of income (score 1-20). 1 means the lowest 5%, and 20 is the highest 5%. ‡Moderate activity means physical activity with light sweating over 30 minutes in a day. For example, brisk walking, jogging, swimming leisurely, bicycling with light effort, gardening, and some housework such as vacuuming.
BMI, body mass index; DM, diabetes mellitus; GC; gastric cancer; HDL-C, high-density lipoprotein cholesterol; SD, standard deviation.”
There are similar mistakes to correct in the other Supplementary Tables. Please, check all of them.
DISCUSSION
This section is adequate. I have only one suggestion: please, include references from 2022 and 2023.
In the end of this section (lines 309-317), we can read: “In conclusion, high serum HDL-C level reduced a gastric cancer mortality in the two independent cohorts. This favorable effect of higher HDL-C level was also observed in a sub-analysis by sex. The favorable effect of higher HDL-C level was also observed after gastrectomy and endoscopic resection. Hence, HDL-C could be considered in the prognosis of gastric cancer. Further studies assessing the modification effect of HDL-C on gastric cancer mortality are warranted to elucidate the factors that could reduce gastric cancer 314 mortality”.
After this paragraph, we find: “5. Conclusions 316 This section is mandatory, with one or two paragraphs to end the main text”
Please, include the conclusion in the appropriate section.
CONCLUSION
See the above comment.
REFERENCES
As pointed out above, please include more references published in 2022 and 2023.
Author Response
We attached the response file.

Round 2
Reviewer 2 Report
There are a few final points that need to be clarified.
1. Regarding one of the main concerns, the authors should describe when the HDLc was measured, at baseline? during the follow-up period?
2. Please review the sentence:
HDL-C levels has an inverse association between HDL-C and cancer risk such as breast7 and gastric cancer.8
I suggest to use:
HDL-C levels has been shown an inverse association with breast7 and gastric cancer.8
3. There are several sentences that should be modified using a correct English and avoiding the use of “effect”, using “association” instead:
In sensitivity analysis among stage I, the effect direction and size was similar with those in overall stage in men, but the inverse association was stronger in women in the analysis of stage I cases than in the overall stage analysis.
Sensitivity analysis showed that the direction and size of the association between HDL-C and death risk in gastric cancer patients with stage I was similar with those in overall stage in men, but the inverse association was stronger in women in the analysis of stage I cases than in the overall stage analysis. In subgroup analysis by treatment, the effect direction and size was similar in overall stage analysis and stage I analysis.
A final review for English style should be also taken into account.
Author Response
There are a few final points that need to be clarified.
- Regarding one of the main concerns, the authors should describe when the HDLc was measured, at baseline? during the follow-up period?
Answer> Thank you for the kind comment. We meaured all epidemiologic and laboratory test including HDL-C at baseline. We clarified this.
Patients with an unknown prognosis (n = 3) and those without data on HDL-C levels at the time of diagnosis (n = 2,628) were excluded.
Table 1. Baseline characteristics of gastric cancer patients by baseline HDL-C (NHISS cohort)
- Please review the sentence:
HDL-C levels has an inverse association between HDL-C and cancer risk such as breast7and gastric cancer.8
I suggest to use:
HDL-C levels has been shown an inverse association with breast7and gastric cancer.8
Answer> Thank you for the kind comment. We revised the sentence as reviewer's comment.
- There are several sentences that should be modified using a correct English and avoiding the use of “effect”, using “association” instead:
In sensitivity analysis among stage I, the effect direction and size was similar with those in overall stage in men, but the inverse association was stronger in women in the analysis of stage I cases than in the overall stage analysis.
Sensitivity analysis showed that the direction and size of the association between HDL-C and death risk in gastric cancer patients with stage I was similar with those in overall stage in men, but the inverse association was stronger in women in the analysis of stage I cases than in the overall stage analysis. In subgroup analysis by treatment, the effect direction and size was similar in overall stage analysis and stage I analysis.
Answer> Thank you for the kind comment. We revised the manuscript.
Sensitivity analysis showed that the direction and size of the association between HDL-C and death risk in gastric cancer patients with stage I was similar with those in overall stage in men, but the inverse association was stronger in women in the analysis of stage I cases than in the overall stage analysis. The effect direction of BMI on death risk (inverted J shape) was similar in both overall stage and stage I, however the hazard ratio was larger in stage I analysis than in overall stage analysis (Supplementary table 7 and Table 3). In subgroup analysis by treatment, the effect direction and size was similar in overall stage analysis and stage I analysis (Supplementary Table 8 and Table 5).
Reviewer 4 Report
Dear authors,
Thank you for performing almost all suggested corrections.
Please replace throughout the text HDL-C ("c" in upper case) with HDL-c ("c" must be in lower case).
In the conclusions, we can read "High serum HDL-C level reduced a gastric cancer mortality in the two independent cohorts. This favorable effect of higher HDL-C level was also observed in a sub-analysis by sex. The favorable effect of higher HDL- C level was also observed after gastrectomy and endoscopic resection.Hence, HDL-C could be considered in the prognosis of gastric cancer.Further studies assessing the modification effect of HDL-C on gastric cancer mortality are warranted to elucidate the factors that could reduce gastric cancer mortality."
In a few lines, HDL-C appears 5 times. Please, rephrase. There are also some grammatical errors.
For example, the authors can use "High serum HDL-c levels reduced gastric cancer mortality in the two independent cohorts. This favorable effect was also observed in an observed sub-analysis by sex. The favorable effect of higher HDL-c levels was also after gastrectomy and endoscopic resection. Hence, this lipoprotein could be considered in the prognosis of gastric cancer. Further studies assessing the modification effect of HDL-c on gastric cancer mortality are warranted to elucidate the factors that could reduce gastric cancer mortality."
Author Response
Thank you for performing almost all suggested corrections.
Please replace throughout the text HDL-C ("c" in upper case) with HDL-c ("c" must be in lower case).
Answer> Thank you for the kind comment.
However, many published articles write HDL-C (upper case). If HDL-c (lower case) is the policy of Journal “Cancers”, we will change overally.
https://pubmed.ncbi.nlm.nih.gov/35843172/
https://pubmed.ncbi.nlm.nih.gov/35583863/
In the conclusions, we can read "High serum HDL-C level reduced a gastric cancer mortality in the two independent cohorts. This favorable effect of higher HDL-C level was also observed in a sub-analysis by sex. The favorable effect of higher HDL- C level was also observed after gastrectomy and endoscopic resection.Hence, HDL-C could be considered in the prognosis of gastric cancer.Further studies assessing the modification effect of HDL-C on gastric cancer mortality are warranted to elucidate the factors that could reduce gastric cancer mortality."
In a few lines, HDL-C appears 5 times. Please, rephrase. There are also some grammatical errors.
For example, the authors can use "High serum HDL-c levels reduced gastric cancer mortality in the two independent cohorts. This favorable effect was also observed in an observed sub-analysis by sex. The favorable effect of higher HDL-c levels was also after gastrectomy and endoscopic resection. Hence, this lipoprotein could be considered in the prognosis of gastric cancer. Further studies assessing the modification effect of HDL-c on gastric cancer mortality are warranted to elucidate the factors that could reduce gastric cancer mortality."
Answer> Thank you for the kind comment. We revised the manuscript as reviewer's comment.
High serum HDL-c levels reduced gastric cancer mortality in the two independent cohorts. This favorable effect was also observed in an observed sub-analysis by sex. The favorable effect of higher HDL-c levels was also after gastrectomy and endoscopic resection. Hence, this lipoprotein could be considered in the prognosis of gastric cancer. Further studies assessing the modification effect of HDL-c on gastric cancer mortality are warranted to elucidate the factors that could reduce gastric cancer mortality.